# Factors affecting the number of influenza patients before and during COVID-19 pandemic, Thailand

Oiythip Yasopa[1], Nontiya Homkham[2], Pornthip Chompook[2]*

1 Department of Disease Control, Division of Epidemiology, Ministry of Public Health, Nonthaburi, Thailand,
2 Faculty of Public Health, Thammasat University, Pathumthani, Thailand

* pchompook@gmail.com

## Abstract

This study was aimed to explore the association between potential factors including public health and social measures and the number of influenza patients in Thailand between 2014–2021. Secondary data from relevant agencies were collected. Generalized Estimating Equation (GEE) and regression coefficient ($\beta$) were performed at a significance level of 0.05. We found factors associated with number of influenza patients during the time prior to COVID-19 pandemic were monthly income per household (Adjusted $\beta$ = -0.02; 95% CI: -0.03, -0.01), population density (Adjusted $\beta$ = 1.00; 95% CI: 0.82, 1.18), rainy season (Adjusted $\beta$ = 137.15; 95% CI: 86.17, 188.13) and winter time (Adjusted $\beta$ = 56.46; 95% CI: 3.21, 109.71). During the time of COVID-19 pandemic, population density (Adjusted $\beta$ = 0.20; 95% CI: 0.15, 0.26), rainy season (Adjusted $\beta$ = -164.23; 95% CI: -229.93, -98.52), winter time (Adjusted $\beta$ = 61.06; 95% CI: 0.71, 121.41), public health control measures (prohibition of entering to into an area with high number of COVID-19 infections (Adjusted $\beta$ = -169.34; 95% CI: -233.52, -105.16), and restriction of travelling also reduced the number of influenza patients (Adjusted $\beta$ = -66.88; 95% CI: -125.15, -8.62) were associated with number of influenza patients. This study commends strategies in monitoring influenza patients to focus on the areas with low income, high population density, and in specific seasons. Public health and social measures which can be implemented are prohibition of entering to risk-areas (lock down), and restriction of travelling across provinces which their effectiveness in reducing influenza infections.

## Introduction

Influenza is an acute respiratory infection which caused many outbreaks around the world and several pandemics with high morbidity and mortality in the past [1]. Influenza is classified into 4 strains (A, B, C and D) and can circulate all through the year [2]. Influenza infections occur among all age groups. It was estimated that more than a billion cases occurred worldwide during year 2019–2030 consisting of 3–5 million cases of severe influenza, and 290,000–650,000 deaths. and several strategic plans were prepared for responding to the influenza

**Competing interests:** The authors have declared that no competing interests exist.

outbreaks which might have occurred anytime [3]. In Thailand, the new influenza A strains (H1N1) 2009 (Influenza A novel H1N1: Pandemic strain) emerged in 2009 which was similar to global situation. There were 120,400 reported cases with the morbidity of 189.73 per 100,000. A total of 30,956 cases were confirmed as the new influenza A (H1N1) 2009 accounting for 48.78 cases per 100,000, mortality rate of 0.31 per 100,000 population, and case fatality rate of 0.64% [4]. Morbidity rate had gradually decreased until year 2014, however, it was increasing since then to the year 2019 showed morbidity rate of 596.16 per 100 000 population and mortality rate of 0.045 per 100,000 population [5]. In 2020, influenza cases had decreased during Coronavirus Disease 2019 (COVID-19) pandemic showing morbidity rate of 186.82 per 100 000 population and case fatality rate of 0.002% [6]. In Thailand, influenza patients occur all year round annually and distributed across the whole country. Number of influenza patients varies by province. The significant factors which related to spreading of the disease showed a different profile in each province. The potential factors associated with the number of influenza patients were seasonal variation, socioeconomic factors, and public health and social measures. This study was aimed to reveal the relationship between these factors and number of influenza patients in Thailand between year 2014 and 2021 (prior and during COVID-19 pandemic).

## Material and methods

### Study design

This is a retrospective longitudinal analytical study using secondary data from 1 January 2014 until 30 June 2021 which covering 77 provinces as the whole country of Thailand.

### Data sources and variables

We collected data from Thai meteorological Department, Thailand national statistics office, and Epidemiology Division, Department of Disease Control for climatic data (1–5), socioeconomic data (6), and number of Influenza cases (7) respectively. Furthermore, population by the area were calculated by using the data from Strategy and Planning Division (8), Ministry of Public Health, and Energy Policy and Planning Office (9), Ministry of Energy. Government control measure regulation by each province (10) were retrieved through Ministry of Interior website. All information of these data were assessed during the dates in 2021 as shown (Table in S1 Table).

Table S1 Data retrieved and its source (S1 Table).

### Influenza notification system

National disease notification system in Thailand notifies several diseases including influenza. All influenza patients are required to be notified through national surveillance system (Report 506) [7]. Health facilities are requested to report any suspected influenza patients through this system. Most government hospitals reported influenza cases daily, however private clinics and hospital rarely reported the cases. Department of Disease control issued the guideline for reporting influenza patients, and International Classification of Diseases (ICD-10): code J10-J11 [8] are advised to follow in reporting an influenza patient. Some patients were further examined by sending their specimens for laboratory confirmation. All probable influenza cases and confirmed cases were diagnosed by using Influenza rapid test and Polymerase Chain Reaction (PCR), respectively.

## Data management and analysis

Each data was collected and cleaned in Microsoft Excel program, then were combined in one file for further analysis. Annual population density per square kilometer by province calculated by mid-year population density and total area in each province using the same data for each month of the same year. Univariable and multivariable generalized estimating equation (GEE) models were used to assess the relationships between related variables and numbers of influenza patients over time. All relevant variables or covariates from univariable analyses with $p$-value $< 0.15$ were included in the multivariable model [9]. A backward model selection procedure was used to identify the multivariable with the best fit. Coefficients regression was used to control for other variables (Adjusted $\beta$) with statistical level of 0.05. The analysis was divided into 2 phases: 1) the period prior to coronavirus (COVID-19) pandemic, during A.D. 2014–2019, and 2) the period of COVID-19 pandemic during A.D. 2020–2021. All analyses were performed with Stata 15.0 (Stata Corporation, College Station, TX, USA).

## Ethical approval

This study obtained ethical approval with research exemption from Human Research Ethics Committee (Science), Thammasat University (COE No. 017/2564). All data were fully anonymized before they were accessed, and the Human Research Ethics Committee waived the requirement for informed consent (Research with Exemption). Other secondary data available on public websites were asked for institutional permission, and all fully anonymous.

## Results

### General characteristics of the study population

All 1,236,299 influenza patients were found between 1 January 2014 until 30 June 2021, consisting of 609,568 males and 626,731 females. Male to female ratio was 1:1.03, mean age was 22.51 ± 20.89 years (max–min = 110 years– 1 month). Most patients were among children under 5 years of age accounting for 21.26%, followed by children between 5–9 years of age (18.91%), and adults: 25–34 years (11.33%). Most of them were Thai citizen (97.82%). More than half lived in municipal area (51.69%), and the rest stayed outside municipal area (48.31%). Most of them sought treatment at clinics or private hospitals (38.08%), followed by community hospitals (33.66%), and general hospitals (18.11%). A total of 279,442 patients were hospitalized accounting for 22.60%, and 296 deaths (0.02%) were reported from all influenza patients.

### Influenza morbidity rates and laboratory confirmation

During the study period, influenza morbidity rate reached the highest point in year 2019 (606 per 100,000 population), followed by 308.99 per 100,000 (in year 2017), and 282.09 per 100,000 (in year 2018). Morbidity rates were lowest during COVID-19 pandemic accounting for 189.14 and 13 per 100,000 in year 2020 and 2021 (only first half of the year) respectively. Influenza laboratory confirmation varied from 21.54% and 33.56% during 2014–2019, however, small proportion of influenza infections were confirmed accounting for 9.24% and 0.12% in 2020 and 2021 respectively. Mostly laboratory confirmation for influenza A virus and influenza B virus distributed a half proportion among them during normal influenza season, whereas influenza A virus showed its predominant of 96% during COVID-19 in 2020 (Fig 1).

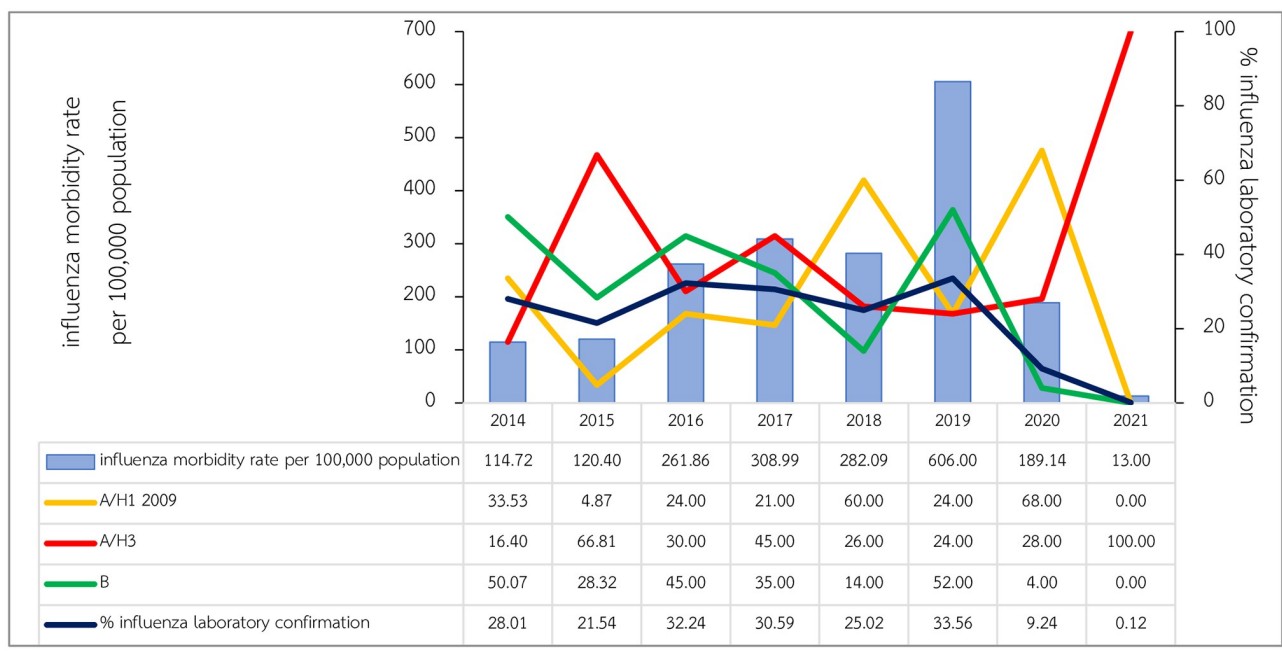

**Fig 1. Influenza morbidity rates in Thailand during 2014–2021.**

## Factors associated with number of influenza patients prior to COVID-19 (2014–2019)

Climatic variables showed strong associations with number of influenza patients in univariable analysis. However, there was no association between average monthly temperature with number of influenza patients ($p$-value = 0.366). There were strong associations between average monthly income per household, population density per square meter, and seasonality with the number of influenza patients ($p$-value <0.001). The people who resided in the province with lower monthly income per household showed an increased number of influenza patients found in that province compared to provinces with more income (Adjusted $\beta$ [adj. $\beta$] = -0.02, 95% CI: -0.03, -0.01; $p$-value <0.001). Population density in each province also showed strong association with number of influenza patients (adj. $\beta$ = 1.00; 95% CI: 0.82, 1.18; $p$-value <0.001) showing an increasing number of influenza patients in a higher population density. Seasonality showed an increasing number of influenza patients in rainy season (adj. $\beta$ = 137.15; 95%CI: 86.17, 188.13) and winter (adj. $\beta$ = 56.46; 95% CI: 3.21, 109.71) compared to summertime with strong significance (Table in S2 Table).

S2 Table Factors associated with number of influenza patients during 2014–2019 (S2 Table).

Monthly income per household by province was available in 2015, 2017, and 2019 following the National Statistics Office protocol which performing the routine national consensus at every other year, which could be included in the analysis. In year 2015, the min-max-average monthly income per household were 13,497.20–45,571.70–23,542.27 Thai Baht (THB). In year 2017, min-max-average monthly income per household were 11,808.90–45,707.30–23,839.78 THB. Finally, in year 2019, min-max-average monthly income per household were 13,970.90–46,977.70–23,567.69 THB, respectively.

### Factors associated with number of influenza patients during COVID-19 (2020–2021)

Population density strongly associated with the number of influenza patients (adj. $\beta$ = 0.20; 95% CI: 0.15, 0.26; $p$-value <0.001). The seasonality was strongly associated with the number of influenza patients ($p$-value <0.001). Number of influenza patients decreased in rainy season (adj. $\beta$ = -164.23; 95% CI: -229.93, -98.52), and the number of influenza patients increased in winter (adj. $\beta$ = 61.06; 95% CI: 0.71, 121.41) which compared to patients in summer. The control measure which showed the maximum reducing number of influenza patients was prohibiton of group gathering in univariable analysis (adj. $\beta$ = -183.02; 95% CI: -229.83, -136.21; $p$-value <0.001), however, this association was not exist in mutivariable analysis. The strong control measures were prohibition of entering to high risk areas (lockdown) (adj. $\beta$ = -169.34; 95% CI: -233.52, -105.16; $p$-value <0.001), followed by retriction of travelling across provinces (adj. $\beta$ = -66.88; 95% CI: -125.15, -8.62; $p$-value = 0.024). Other public health control measures showed their effects on reducing number of influenza patients in univariable analysis, however these associations were not exist in multivariable analysis (Table in S3 Table).

S3 Table Factors associated with number of influenza patients during 2020–2021 (S3 Table).

## Discussion

This study shows influenza morbidity during COVID-19 pandemic was extraordinary from its regular season before COVID-19 which were similarly observed globally [10–12]. Even though, our finding showed a low number of influenza infections in 2020, but it was not at historically low of its peak during the time before COVID-19. This phenomenon could be explained by the effects of non-pharmaceutical interventions, and other implementations in order to control COVID-19 transmission, and reduce mortality rate. Influenza strains confirmed in this study were conclusive to those which were regularly tested during normal influenza season [5].

This study demonstrated socioeconomic status, population density, and seasonal variation associated with the number of influenza patients in Thailand. Socioeconomic status plays a crucial role in public health. It was found the area with high transmission and mortality rates were among people with low income [13], and 2.8 times more likely that influenza-like illness to be found than those with high socioeconomic status [14]. Moreover, the increasing risks of other infections of *Plasmodium falciparum*, hookworm, *Entamoeba histolytica*, and *Entamoeba dispar* were also found in the areas among people with low socioeconomic status [15]. The evidence of health outcomes as the consequences of socioeconomic status was overwhelmed [16]. In other words, when monthly income increased, it would lead to better health behaviors including residential features to protect oneself from disease infections [17]. Regarding population density, influenza morbidity rates had increased with an increasing population density in the United States [18]. Moreover, influenza epidemic in India during 1918–1919 found that population density below 175 people per square mile had death rate of 3.72%, whereas in the areas more than 175 people per square mile showed the death rate of 4.69% [19]. These show population density as an important factor on increasing the ability of influenza transmission among crowded people.

Evidence of seasonality associated with number of influenza patients was found similarly to other studies. In Thailand, during influenza pandemic in 2009, number of influenza patients reached its top in rainy season followed by winter, and showed the lowest number in summer [20]. Another study from Lebanon also found the same pattern of respiratory viral infections showing the highest number of patients in the rainy season, and lowest in summer [21]. It was

found that influenza outbreaks normally occurred in rainy season [22]. However, the number of influenza patients were mostly found in winter and summer and reached the lowest number in rainy season during COVID-19 pandemic in 2020–2021. This was a phenomenon showing a different pattern during the time before COVID-19 pandemic (2014–2019). This unusual pattern of influenza infections found in this study also occurred similarly elsewhere during the COVID-19 pandemic in the United States [10], and Bangladesh [11] where seasonal influenza patients had occurred less number than in its normal season. This can be explained by all intense control measures during COVID-19 pandemic which could reduce the number of influenza infections and other viral respiratory infections as well as COVID-19. Our study could not find the association of monthly rainfall and number of influenza patients in multi-variable analysis which was similar to another Thai study [20].

The evidence of public health and social measures (PHSMs) previously known as non-pharmaceutical interventions (NPIs) [23] showed its effects on reducing number of seasonal influenza in several tropical countries as countermeasures on COVID-19 [24]. Prohibition of travelling to restricted areas (lock down) was a significant control measures associated with reducing the number of influenza patients found in this study. It was also found that limitation of travelling could reduce the occurrence of influenza up to 41%, and other respiratory infections in nine tropical Asian countries [24]. Locking down the area was found as an effective countermeasure to control COVID-19 as its aim [25,26], however it may be uneasy to be implemented in another country [27]. Moreover, travel restriction at specific times (curfew) also showed a significant correlation in reducing the number of influenza patients. The evidence of the restriction of travelling showed its effect on COVID-19 transmission by delaying the peak of epidemic by 2 weeks and reducing the number of patients up to 33% in China [28]. Moreover, the evidence showed the number of influenza patients decreasing dramatically during control measures on COVID-19 elsewhere [29]. Unfortunately, we could not find other public health measures associated with number of influenza patients, however, univariable analysis showed some degree of reducing number of influenza infections apart from hygiene etiquette mostly done by Thai people including other infrastructure during COVID-19 [30].

This study may have limitations. Firstly, our findings based on the secondary data of reported influenza patients who were not totally laboratory confirmed for influenza infections. However, the national disease notification system (Report 506) and ICD-10 system would allow most influenza cases to be reported correctly. Moreover, influenza season found in this study was consistent with seasonal influenza occurred annually which could support evidence most influenza patients were included in this study. Secondly, an unusual low number of reported influenza patients during COVID-19 (2020–2021) was consistent with other findings. However, the small number of influenza patients might have caused the undetectable association between potential variables, like school closures, or prohibition on group gatherings with the number of influenza patients in multivariable analysis. The changing health care seeking behaviours among people, and burden of heath care providers could be the other reasons of decreasing number of influenza notification during COVID-19. Thirdly, the monthly income data at each province was unavailable during COVID-19 (2020–2021), hence this factor could not be included in the analysis. However, monthly income shows its crucial relationship with the number of influenza patients prior the pandemic.

## Conclusion

Our study recommends policy implication on surveillance system and controlling influenza infections should focus on the proxy variables for socioeconomic status which are low-income area, or high population density. Further intervention can be varied upon seasonality,

specifically during rainy season and winter. Public health and social measures showed strong evidence in reducing number of influenza infections, specifically with prohibition of travelling to restricted area (lock down), and travel restriction at specific time (curfew) could reduce the number of influenza infections including other respiratory infectious diseases, perhaps to better prepare for the next influenza pandemic.

## Supporting information

**S1 Table. Data retrieved and its source.**
(PDF)

**S2 Table. Factors associated with number of influenza patients during 2014–2019.**
(PDF)

**S3 Table. Factors associated with number of influenza patients during 2020–2021.**
(PDF)

**S1 Data.**
(XLSX)

## Acknowledgments

We would like to thank the Department of Disease Control, Meteorological Department, and National Statistical Office for providing their data to this study. Our thanks also go to the Strategy and Planning Division of Ministry of Public Health, Energy Policy and Planning Office, and Center for COVID-19 Situation Administration for all essential information from their websites contributing to this study.

## Author Contributions

**Conceptualization:** Oiythip Yasopa, Nontiya Homkham, Pornthip Chompook.

**Data curation:** Oiythip Yasopa, Nontiya Homkham.

**Formal analysis:** Oiythip Yasopa, Nontiya Homkham.

**Methodology:** Nontiya Homkham, Pornthip Chompook.

**Supervision:** Pornthip Chompook.

**Writing – original draft:** Oiythip Yasopa, Pornthip Chompook.

**Writing – review & editing:** Oiythip Yasopa, Nontiya Homkham, Pornthip Chompook.

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
