## [Decision Letter · Decision Letter 0]

19 Mar 2024

PONE-D-23-31302Factors affecting the number of influenza patients before and during COVID-19 pandemic, ThailandPLOS ONE

Dear Dr. Chompook,

Thank you for submitting your manuscript to PLOS ONE. After careful consideration, we feel that it has merit but does not fully meet PLOS ONE’s publication criteria as it currently stands. Therefore, we invite you to submit a revised version of the manuscript that addresses the points raised during the review process.

**ACADEMIC EDITOR: Kindly check the comments made by the Reviewer (Reviewer 2) and attend to the areas highlighted.**

We look forward to receiving your revised manuscript.

Kind regards,

Olatunji Matthew Kolawole, Ph.D.

Academic Editor

PLOS ONE

Journal Requirements:

Reviewers' comments:

Reviewer's Responses to Questions

**Comments to the Author**

1. Is the manuscript technically sound, and do the data support the conclusions?

Reviewer #1: Yes

Reviewer #2: Partly

2. Has the statistical analysis been performed appropriately and rigorously? 

Reviewer #1: Yes

Reviewer #2: I Don't Know

3. Have the authors made all data underlying the findings in their manuscript fully available?

Reviewer #1: Yes

Reviewer #2: Yes

4. Is the manuscript presented in an intelligible fashion and written in standard English?

Reviewer #1: Yes

Reviewer #2: Yes

5. Review Comments to the Author

Reviewer #1: This article provides valuable insights into the factors associated with the number of influenza patients in Thailand before and during the COVID-19 pandemic. The study uses secondary data from relevant agencies and employs statistical models to assess the relationships between related variables and the number of influenza patients over time. The article is well-structured and provides detailed information on the study design, data sources, and statistical methods used to analyze the data. The findings suggest that factors such as monthly income per household, population density, and seasonality are associated with the number of influenza patients in Thailand. The study also recommends policy implications for surveillance systems and controlling influenza infections, with a focus on socioeconomic status and public health and social measures. The English in the article appears to be well-written and technically sound, with appropriate use of scientific terminology and clear presentation of the study findings.Overall, this article provides important information for researchers, policymakers, and healthcare professionals interested in understanding and addressing the factors affecting influenza infections in Thailand.

Reviewer #2: 1) The study doesn't seem replicable

There is need to define the range for the level of income in the data/result

Suggest, a population of each province be made available to get a better picture of the cases and also

The influenzas cases were confirmed using what diagnostic method? needs to be stated

6. PLOS authors have the option to publish the peer review history of their article (what does this mean?). If published, this will include your full peer review and any attached files.

Reviewer #1: No

Reviewer #2: No

---

## [Author Response · Author response to Decision Letter 0]

10 Apr 2024

Dear Reviewers,

Reviewer#1: Thank you so much for your kind support to our study as shown in your precious comments to the Author.

Reviewer#2: Thank you so much for your precious comments indicating that the manuscript needs further clarification.

Dear Editors,

Thank you so much for your kind support.

all the very best,

Pornthip Chompook

---

## [Editor Report · Decision Letter 1]

24 Apr 2024

Factors affecting the number of influenza patients before and during COVID-19 pandemic, Thailand

PONE-D-23-31302R1

Dear Dr. Chompook,

We’re pleased to inform you that your manuscript has been judged scientifically suitable for publication and will be formally accepted for publication once it meets all outstanding technical requirements.

Kind regards,

Olatunji Matthew Kolawole, Ph.D.

Academic Editor

PLOS ONE
---

## [Editor Report · Acceptance letter]

29 Apr 2024

PONE-D-23-31302R1 

PLOS ONE

Dear Dr. Chompook, 

I'm pleased to inform you that your manuscript has been deemed suitable for publication in PLOS ONE. Congratulations! Your manuscript is now being handed over to our production team.

Kind regards, 

on behalf of

Dr. Olatunji Matthew Kolawole 

Academic Editor

PLOS ONE